# Chromatographic Applications in the Multi-Way Calibration Field

**DOI:** 10.3390/molecules26216357

**Published:** 2021-10-21

**Authors:** Fabricio A. Chiappini, Mirta R. Alcaraz, Graciela M. Escandar, Héctor C. Goicoechea, Alejandro C. Olivieri

**Affiliations:** 1Laboratorio de Desarrollo Analítico y Quimiometría (LADAQ), Cátedra de Química Analítica I, Facultad de Bioquímica y Ciencias Biológicas, Universidad Nacional del Litoral, Ciudad Universitaria, Santa Fe S3000ZAA, Argentina; fabriciochiappini@gmail.com (F.A.C.); alcarazmirtaraquel@gmail.com (M.R.A.); hgoico@fbcb.unl.edu.ar (H.C.G.); 2Consejo Nacional de Investigaciones Científicas y Técnicas (CONICET), Godoy Cruz C1425FQB, Argentina; escandar@iquir-conicet.gov.ar; 3Departamento de Química Analítica, Facultad de Ciencias Bioquímicas y Farmacéuticas, Universidad Nacional de Rosario, Instituto de Química de Rosario (IQUIR-CONICET), Suipacha 531, Rosario S2002LRK, Argentina

**Keywords:** liquid and gas chromatography, multi-way chromatographic data generation, multi-way models and calibration, analytical figures of merit

## Abstract

In this review, recent advances and applications using multi-way calibration protocols based on the processing of multi-dimensional chromatographic data are discussed. We first describe the various modes in which multi-way chromatographic data sets can be generated, including some important characteristics that should be taken into account for the selection of an adequate data processing model. We then discuss the different manners in which the collected instrumental data can be arranged, and the most usually applied models and algorithms for the decomposition of the data arrays. The latter activity leads to the estimation of surrogate variables (scores), useful for analyte quantitation in the presence of uncalibrated interferences, achieving the second-order advantage. Recent experimental reports based on multi-way liquid and gas chromatographic data are then reviewed. Finally, analytical figures of merit that should always accompany quantitative calibration reports are described.

## 1. Introduction

In the last decades, the use of chromatographic techniques coupled to multidimensional detection systems has gained the attention of analytical chemists. This is due to the fact that its combination with multi-way calibration tools allows one to obtain a large amount of chemical information and resolve highly complex systems outperforming the advantages of classical univariate chromatographic methods [1,2,3]. It has been demonstrated that the use of multi-way chromatographic data analysis yields better analytical performance than those based on univariate calibration, achieving additional analytical benefits, such as reducing the time of analysis, decreasing the solvent consumption and avoiding sample pre-processing steps. All these characteristics are consequences of the fact that if properly processed, multi-way data allow one to achieve the well-known second- and third-order advantages [4].

In the context of higher-order calibration methods (second-, third- and higher-order), the concept of the second-order advantage relies on the fact that, in certain circumstances, the contribution of individual sample constituents can be accurately obtained, even in the presence of unmodeled or unexpected interferences [5]. This property has been extensively demonstrated in a wide variety of applications, as can be verified in a vast number of publications. Even though this property was first proposed for second-order/three-way calibration methods, it has been extended to higher-order calibration methodologies. However, in the case of third- or higher-order data (four-way calibration and beyond), the additional advantages are still in discussion, and there is no general consensus about the real nature of the third-order advantage. Nevertheless, experimental and theoretical work are continuously growing in order to investigate the benefits associated to the increment in the number of instrumental data modes [6]. 

Second-order data consists in a collection of a bidimensional data array for a given sample. The bidimensional signals acquired for a set of samples can then be joined to obtain a three-way data array. This tensorial object is characterized by three experimental modes, one corresponding to the number of samples and the additional ones representing the measured analytical information. In particular, second-order data are generated when the analytical signals are recorded in two independent instrumental modes. In this regard, chromatography with spectral detection represents the most reported analytical application in the second-order calibration field. For instance, liquid chromatography with spectroscopic diode array detection (LC-DAD), fast scanning fluorescence detection (LC-FSFD), spectral mass spectrometric detection (LC–MS) and gas chromatography with spectral mass spectrometric detection (GC–MS) allow one to obtain second-order data.

On the other hand, third-order data are obtained when an additional experimental or instrumental mode is incorporated. In principle, there is no theoretical limitation to the benefits that may be brought about by additional instrumental modes in higher-order calibration [7]. Nevertheless, there might be a limitation for the generation of multidimensional data, since most multi-way chemometric modelling approaches imply strong assumptions about the mathematical structure of the data arrays. This restraint may be apparently overcome by virtue of the development of novel instrumental technology. 

Even though countless alternatives would be possible for implementing third-order data acquisition, third-order chromatographic methodologies are one of the most challenging approaches. In the first attempt of acquiring third-order chromatographic data, in 1981, Apellof and Davidson reported a chromatographic method with excitation-emission matrices (EEM) detection for qualitative analysis [8]. Since then, interest in this type of data has been growing, accompanied by the development of novel and robust chemometric models capable of exploiting the potentiality of the multi-way data arrays [6], and also by the progress and expansion of analytical instrumentation.

Notwithstanding the wide variety of chemometric models covering a broad range of possibilities, it should be noted that there is no single chemometric model able to fit all the types of data that can be experimentally measured in a laboratory. Among all the reported models, the most used ones in second- and third-order calibration are parallel factor analysis (PARAFAC) [9], multivariate curve resolution-alternating least-squares (MCR-ALS) [10], and partial least-squares-based techniques, followed by residual multilinearization, e.g., unfolded- or multiway- partial least-squares, followed by residual multilinearization (U-PLS/RML and N-PLS/RML) [11]. In addition, variants of the above-mentioned models have been developed, aiming to overcome some limitations and to improve its performance. PARAFAC2 [12], augmented PARAFAC (*A*PARAFAC) [13] and the family of alternating multilinear decomposition models (AMLD) [7] are examples of PARAFAC variants. 

Together with data modelling, another essential part in the development and validation of calibration methodologies concerns the estimation of analytical figures of merit (AFOMs). These figures are numerical parameters used to characterize the performance of a developed protocol and to compare the relative success among different methodologies [14]. In multi-way calibration, the theory of error propagation is the core of the AFOM estimators. This topic has been a focus of recent discussions, and important advances have been reported in the literature [14] regarding the estimation of crucial AFOMs, such as sensitivity, analytical sensitivity, limit of detection and limit of quantitation. 

The present review is intended to provide a comprehensive coverage of the most relevant alternatives reported for the generation and analysis of second- and third-order chromatographic data and their application in the multi-way calibration field. 

## 2. LC Multi-Way Data Generation

### 2.1. Second-Order Data

Second-order data are characterized by the presence of two different instrumental modes in the data array acquired for each experimental sample. From the experimental point of view, there are two conceptually different methods to generate second-order data: (1) using a single instrument and (2) connecting two instruments in tandem. In the former case, measurements are directly performed on a single equipment, either because two components of the instrument itself are able to provide each of the data modes, or because first-order (vectorial) measurements are made as a function of time, and these vectors are joined to produce a data matrix per sample. In the latter, each of the connected instruments provides a data mode or matrix direction to the measured matrix data.

Although not directly related to chromatography, EEM are, probably, the most explored second-order data measured in a single instrument [15], and are good examples to introduce the notion of trilinearity. EEM data are, under certain circumstances, classified as trilinear [16] and, thus, robust and often unique trilinear decomposition models can be applied to them. This is because the excitation and emission component profiles are independent phenomena, and (properly normalized) do not depend on the sample. However, second-order fluorescence data may not always be trilinear [16]. For example, if inner filter effects occur in one of the data modes, lack of trilinearity will be observed, because the fluorescence profile of a given component will be different across samples in the mode where the inner filter takes place [17]. This is due to the fact that the magnitude of the inner filter depends on the concentration of the constituent producing the effect. These data will be classified as non-trilinear type 1 if the inner filter affects a single data mode, and type 2 if it affects both instrumental modes [16]. 

Briefly, non-trilinear (and non-multilinear) data types are those where the multi-linearity is lost because either (1) component profiles are not constant along one or more modes or (2) there is mutual dependence between the phenomena taking place in the instrumental modes. Specifically, three-wat data are non-trilinear type 1 if there is a single trilinearity breaking mode (non-constant profiles along this mode), non-trilinear type 2 if there are two breaking modes and non-trilinear type 3 if there is mutual dependence between the two instrumental modes. 

The connection of two instruments in tandem provides another convenient way of generating second-order data. In fact, chromatography with multivariate spectral detection shares with matrix fluorescence data the priority in the publication record regarding second-order calibration protocols. Detection using a diode array detector (DAD), a fast-scanning fluorescence detector (FLD) or a spectral mass spectrometer (MS) provides the spectral mode to a liquid chromatograph, which itself is responsible for the elution time mode of the measured data. In the case of gas chromatography, spectral MS detection is the method of choice for generating second-order data [1]. In a few cases, electrophoretic measurements have replaced the chromatographic separation mode [18]. 

All data stemming from chromatographic measurements should be considered, in principle, non-multilinear type 1 [2]. This is because the reproducibility across sample injections, both in the position and shape of chromatographic peaks, is never perfect. However, if the elution time mode is indeed reproducible or quasi-reproducible, either because the total experimental time of each run is short enough, or because the instrument itself provides reproducible data, e.g., a gas chromatograph, then the data could be considered trilinear [19].

### 2.2. Third-Order Data

As an extension of second-order, third-order data are characterized by the existence of three instrumental modes in the data that are collected for a given sample. In the chromatography field, third-order data are usually obtained through the hyphenation of two different instrument, i.e., a chromatograph coupled to a second-order detector (for example, EEM), or by means of a two-dimensional chromatographic instrument with vectorial signal detector, e.g., DAD or FLD, among others [20]. 

Literature reports regarding third-order data analysis are still scarce in comparison with second-order calibration methods for chromatographic applications. This fact could be a consequence of the complexity associated to the required instrumental arrangements and the intrinsic difficulties of monitoring in-flow analysis. 

In time-dependent experiments, the sample composition changes with the time, which is monitored when registering the analytical signal. In these kinds of systems, measurements can be conducted by performing different experiments, either in steady-state conditions or in continuous-flow conditions. In steady-state flow applications, the system is monitored at different periods of time in a condition where it no longer evolves, and any property associated with the flow or time remains constant. Stopped-flow systems and fraction collection-based methodologies are examples of steady-state flow applications. Under these circumstances, the instrumental modes of the third-order data are fully independent between them. On the contrary, in continuous-flow systems, the first issue to be considered is the synchronization between the evolving rate of the system and the scan rate of the detector. If full synchronization exists, the instrumental modes are mutually independent and the data will fulfil one of the multilinearity conditions [16]. For instance, DAD systems coupled to LC allow acquiring an entire spectrum at every chromatographic time and then, bilinear second-order data (elution time × spectra) are obtained. However, for detectors based on spectral scanning, the synchronization issue is not trivial. 

The most reported strategy for generating third-order data consists in the acquisition of EEM as a function of the chromatographic time (LC-EEM). To generate these kinds of data, several instrumental arrangements have been proposed. In this regard, it should be reminded that a fluorescence spectrometer is a second-order instrument that enables the acquisition of bidimensional arrays as a function of elution time. Nevertheless, the commercially available spectrofluorometers operate through mechanical motion of the gratings to generate complete spectra or EEMs. These motions demand a finite time, which is considerably larger than the one required to benchmark the evolving rate of the chromatographic system. Despite the fact that modern analytical instrumentation has simplified the generation of multi-way data, third-order chromatographic acquisition is still a challenging task from the instrumental standpoint, and also constitutes a challenge from the chemometric perspective.

A number of analytical methodologies based on the generation of third-order LC data using fluorescence detection have been reported. One of the strategies follows the path pioneered by Apellof and Davidson [8] who proposed the generation of third-order LC data with qualitative aims by acquiring EEM at discrete chromatographic times. This strategy is based on the collection of discrete fractions eluting from the chromatograph for which an EEM is then obtained using a conventional spectrofluorometer. In 1997, R. Bro [21] proposed, for the first time, an approach for the generation of third-order LC-EEM data with quantitative aims based on Apellof and Davidson’s idea. Samples were analysed under identical conditions and a four-way array was then built and subjected to chemometric decomposition. In 2014, Alcaraz et al. [22] implemented the same strategy for the quantitation of three fluoroquinolones in water samples using a custom-made fraction collector, which was connected to the end of the chromatographic column and enabled the collection of fractions in a 96 wells-ELISA plate. At the end of the sampling, the plate was placed into a conventional spectrofluorometer for registering the EEM of each well. Even though the instrumental modes are mutually independent and synchronization between rates is not demanded by this strategy, several issues are undesired from the chemometric standpoint. Despite the fact that the third-order data obtained for each sample are trilinear, the four-way array built for a set of samples does not behave as quadrilinear, but as non-quadrilinear type 1, because of the lack of time and peak shape reproducibility between runs. Recall that non-quadrilinear four-way data are type 1 for a single breaking mode, type 2 for two breaking modes, type 3 for three breaking modes and type 4 if there is dependence among pairs of instrumental modes.

A different chromatographic approach was first introduced by Muñoz de la Peña’s group [23] and was then implemented in other applications with quantitative purposes [24,25]. In this case, to avoid stopping the flow, the FLD capabilities were exploited and the third-order LC-EEM data were measured by performing several chromatographic runs of aliquots of the same sample. The corresponding second-order LC-FLD data matrix were then registered, changing the excitation wavelength for every injected aliquot. Hence, the three-way array was built by joining the data matrices acquired at each excitation wavelength. This approach is characterized by the fact that a single chromatograph is used for the acquisition of the third-order data, including an autosampler and an FLD. However, deviations of multilinearity are present when this approach is used. The first aspect to be considered is the fact that time shifts, and peak distortions may appear among runs, breaking the trilinearity of the third-order LC-EEM data (the same phenomenon occurring in second-order calibration). Hence, the four-way data array will not be quadrilinear, but non-quadrilinear type 4, which is a complex scenario for chemometric models. To the best of our knowledge, there are no efficient pre-processing tool for recovering the trilinearity of LC-EEM third-order data, and no adequate models dealing with non-quadrilinearity type 4. However, this data can be properly modelled by implementing a bilinear decomposition of a super-augmented bilinear data matrix (see Section 3). It should be noticed that only a small number of runs are performed per sample to reduce time, sample and reagent consumption, leading to imbalanced data arrays with many data points in the emission and the chromatographic directions and only a few points in the excitation direction. These issues may hinder its application for the analysis of complex systems with a large number of constituents.

Finally, the most explored and promising approach is the one consisting in the hyphenation of a chromatograph and a fast-scanning spectrofluorometer connected through a fluorescence flow-cell. The first work that report the implementation of an online EEM registering system was done by Goicoechea’s group [25], who described and analysed the advantages and disadvantages of this alternative in comparison with the two aforementioned ones. This approach presents the great advantage of simultaneously recording the EEM on-line with the LC procedure, achieving a drastic reduction of time analysis, reagents and sample. Notwithstanding, the main drawback is related to the strong dependence of the elution time mode with both spectral modes, which leads to a loss of trilinearity in the third-order data, and to non-quadrilinear data of type 4 for multi-sample analysis. This phenomenon occurs as a consequence of the lag between the elution and the fluorescence scanning rates. To cope with these limitations, researchers have implemented novel instrumental configurations [26], developed new spectrometers [27] and introduced new chemometric alternatives [28]. Escandar’s group implemented a chromatographic setup that decreased the time dependence effect by reducing the linear flow rate of the mobile phase, incorporating a large inner-diameter tube between the column and the flow-cell [26]. In this way, the authors reported that the time-dependence effect is negligible, and the third-order data of individual samples are indeed trilinear. In addition, due to the large chromatographic times and the slow chromatographic rate, reproducibility in the elution time among samples was observed leading to quadrilinear data. More recently, Alcaraz et al. have presented an ultra-fast multi-way detector that enables measuring a complete EEM by bidimensional excitation and emission spatial dispersion [27]. This device, based on the use of a CCD camera, allows acquiring fluorescence images in the order of milliseconds. The first advantage of this setup is the feasibility of acquiring trilinear third-order data with several data points in all the three modes. For instance, three-dimensional arrays of size 1450 × 240 × 320 for elution time, excitation and emission modes, respectively, were reported. In this case, non-quadrilinear type 1 four-way data arrays are generated, which can be easily decomposed by known chemometric models such as *A*PARAFAC, MCR-ALS and U-PLS with residual quadrilinearization (RQL), among others. This device represents a step-forward in the field of third-order data acquisition for dynamic systems.

Totally different third-order chromatographic data can be generated by bidimensional (2D) chromatography in combination with vectorial detection or by three-dimensional (3D) chromatography. In the case of 2D chromatography, a sample is driven through two independent columns; the effluent from the first-dimension column is sequentially injected into the second-dimension column. At the end of the latter, a spectral detector registers a vectorial signal at each chromatographic time. In this way, the generated third-order data involve the first-dimension chromatographic time, the second-dimension injections and the corresponding spectra. In this regard, LC^2^-DAD [29] and GC^2^-MSTOF (time of flight) methodologies have been reported and four-way calibration method have been successfully implemented [30]. More recently, a new methodology based on 3D GC with univariate detection (FID or single quadrupole MS) was proposed as an alternative to yield third-order data with quantitative aims [31]. In this case, the three-dimensional data array is built by the combination of the three chromatographic modes. All these strategies share the same undesired particularity in chemometric terms, i.e., the elution time shifts and peak distortions that occur among samples in both the first- and the second-dimension (and the third one in the case of 3D chromatography), which, in principle, preclude the application of a multilinear model. 

## 3. LC Multi-Way Data Analysis: Chemometric Models and Algorithms

As shown in the previous section, the generation of three- and four-way data implies the construction of a variety of data arrays, characterized by various mathematical properties. Hence, it is not surprising that a large diversity of models and algorithms have proliferated in recent decades. 

The concept of data multilinearity, i.e., bilinearity, trilinearity and quadrilinearity is the common thread in method taxonomy and characterization [16]. These concepts should orientate the analyst in the selection of the most suitable method for a given calibration scenario. In this regard, it is important to consider that despite the structure of the raw experimental data, they can be subjected to mathematical operations prior to chemometric processing, in order to fulfil the conditions required for a successful chemometric decomposition. For instance, second-order matrices can be arranged as follows: (1) they can be stacked in a third mode to give rise to a three-way data array, (2) joined in-plane to produce an augmented data matrix in either of the instrumental directions, or (3) unfolded into vectors and then join the vectors to produce a single matrix. Two key factors determine the selection of any of the latter structures: the existence of phenomena producing constituent profiles which vary from sample to sample, and/or mutually dependent phenomena occurring in the two instrumental modes. The different data arrays that can be obtained from a second-order dataset, prior to chemometric modelling, are illustrated in Figure 1.

Following the same criteria described before, the third-order arrays in four-way calibration can be organized as follows: (1) they can be disposed in a fourth mode yielding a four-way data array; (2) organized in an augmented three-dimensional data array in either of the instrumental directions; (3) joined in-plane to generate a bidimensional data matrix with one or two augmented modes; and (4) completely unfolded into vectors and then stacked in a single matrix. These possibilities are graphically summarized in Figure 2.

In general terms, three families of chemometric modelling approaches can be distinguished for three- and four-way data analysis. Each group of methods is characterized by assuming different hypotheses about the data structure and exhibit different degrees of flexibility. This leads to distinctive model interpretabilities and capabilities of exploiting the benefits of multi-way data arrays in terms of analytical performance and solution uniqueness. Although a vast list of mathematical models and algorithms exists, only the most relevant ones are summarized and detailed in this report. 

In accordance with the different types of required mathematical operations, stacked multi-way data that conform the property of multilinearity can be subjected to multilinear decomposition methods, mainly represented by PARAFAC and AMLD [20], which define the first family of models here described. They are based on a multilinear decomposition procedure, and present the following advantages: (1) no need of special initialization methods, since the solutions are often unique, (2) constraints may not be necessary, in general, to drive the optimization phase to the final solution, (3) figures of merit are well-known and (4) the decomposition of multi-dimensional data in their original structure is known to be more efficient than unfolding the data into arrays of lower dimensions [32]. The uniqueness property which is often achieved in this kind of decomposition is directly tied to the possibility of exploiting the relevant second-order advantage in analytical protocols.

Given a set of I bilinear second-order data with J and K data points in each of the instrumental modes and N responsive constituents, a three-way data arrangement $\underline{X}$ of size I×J×K can be obtained. If the tensorial object $\underline{X}$ obeys the property of low-rank trilinearity, then each element can be expressed as:(1)$$x_{ijk}=\overset{N}{\underset{n=1}{\sum }}a_{in}b_{jn}c_{kn}+e_{ijk}$$
where $a_{in}$, $b_{jn}$ and $c_{kn}$ are the ith, jth and kth elements of the profile matrices $A_{I\times N}$, $B_{J\times N}$ and $C_{K\times N}$ in the nth column vectors, respectively. The scalar $e_{ijk}$ denotes a generic element of the three-way residual array ${\underline{E}}_{I\times J\times K}$. Under this assumption, $\underline{X}$ arrangement can be submitted to trilinear decomposition, according to the following model formulation: (2)$$\underline{X}=\overset{N}{\underset{n=1}{\sum }}a_{n}⊗b_{n}⊗c_{n}+\underline{E}$$
where ⊗ indicates the Kronecker product [9] and **a***_n_*, **b***_n_* and **c***_n_* are the *n*th. columns of **A**, **B** and **C**, respectively.

The same model formulation expressed by Equation (2). can be directly extended to a multilinear multi-way data arrangement of any order. For instance, in the case of third-order data, i.e., four-way calibration, and additional instrumental mode is included. If L data points are registered, then, quadrilinear decomposition follows the expression: (3)$$\underline{X}=\overset{N}{\underset{n=1}{\sum }}a_{n}⊗b_{n}⊗c_{n}⊗d_{n}+\underline{E}$$
where $D_{L\times N}$ contains the profiles of N constituents in the third instrumental mode and **d***_n_* is the *n*th. column of **D**.

For three-way calibration, various optimization models for the estimation of **A**, **B** and **C** matrices have been proposed. In this sense, direct trilinear decomposition method (DTLD) [33] is based on eigenvalue-eigenvector decomposition, whereas PARAFAC [9] and AMLD [7] are based on alternating least-squares philosophy. In particular, AMLD includes a variety of strategies, such as alternating trilinear decomposition (ATLD) [34] or self-weighted ATLD (SWATLD) [35] for second-order data, and alternating penalty quadrilinear decomposition (APQLD) [36] or regularized self-weighted alternating quadrilinear decomposition (RSWAQLD) [37] for third-order data. The main advantage of these models, in general, relies in the fact that the multilinear decomposition is unique. From the analytical point of view, this property implies that, even when initial estimators are unknown, the optimization phase would retrieve the true constituent profiles along each of the instrumental modes, as well as the relative contribution of each component along the sample mode. The latter are then used to build the so-called pseudo-univariate calibration curve for quantitative purposes. 

Notwithstanding, all these benefits can be achieved only if the modes are mutually independent from each other. This is not generally the common rule for chromatographic multi-way data. As stated above, multi-way chromatographic data are normally classified as non-trilinear type 1 because the reproducibility across sample injections is never perfect. In general, multilinear decomposition is the least flexible option, since it is sensitive to the lack of multilinearity [21]. This issue can be overcome by implementing different strategies. In certain cases, the lack of reproducibility in the sample mode can be solved by applying pre-processing procedures for chromatographic peak alignment [38]. Besides, different MLD variants derived from the previously mentioned approaches have also emerged aiming at dealing with the lack of multilinearity in chromatographic data. For instance, PARAFAC2 is a variant of the classical PARAFAC, which is based on an alternative form of Equation (2) [12]: (4)$$x_{ijk}=\overset{N}{\underset{n=1}{\sum }}a_{in}b_{jn}\left(i\right)c_{kn}+e_{ijk}$$
where $b_{jn}\left(i\right)$ is the *j*th. element of the **b***_n_* profile along the sample dependent mode and depends on the sample index i. In addition, to follow Equation (4), the PARAFAC2 model requires constant cross product between all pairs of **b***_n_* profiles. This model formulation allows the chromatographic profiles not to be identical from run to run. For more details the reader must be referred to Ref [12]. Despite that this method proved to have some success in the field of chromatography, it is still a non-flexible model since it assumes that the degree of overlap between chromatographic peaks is constant for each pair of constituents in all samples [12,39]. 

The second group of multi-way calibration methodologies is represented by bidimensional decomposition and curve resolution methods, where MCR-ALS [40] emerges as the most important modelling approach for chromatographic multi-way data. Since its original publication, MCR-ALS has been the subject of extensive research in both fundamental and applied chemometrics. To put it succinctly, in contrast to multilinear decomposition methods, the MCR-ALS approach consists in performing a bilinear decomposition of a bilinear data arrangement X of size J×K, according to:(5)$$X=CS^{T}+E$$
where $C_{J\times N}$ and $S_{K\times N}$ are matrices that capture the pure instrumental responses of N components in each of the instrumental modes, respectively, and $E_{J\times K}$ collects the model residuals. The bilinear decomposition is achieved by virtue of the ALS algorithm. Besides, in contrast to the previously mentioned methods, ALS initialization with random values is not a convenient strategy. On the contrary, MCR-ALS is a more flexible model but suffers from ambiguity phenomena, which can have dramatic effects on the analytical performance of a calibration protocol [41,42]. Hence, in the usual application of this model, initial estimates of C or S are commonly obtained through the so-called purest variables methodology [43]. On the other hand, the study of MCR-ALS ambiguity is still a matter of extensive research in the chemometric field [41]. In general terms, the extent of RA can be mitigated through the incorporation of mathematical and chemically sensible constraints during ALS optimization, such as non-negativity, unimodality, selectivity, correspondence between species, among others [44,45,46,47,48,49,50,51]. 

For the particular case of three-way calibration, the most common approach for chromatographic data modelling is based on generating an augmented data array, i.e., a set of I bilinear matrices X are disposed in a column-wise augmented arrangement $X_{\mathrm{aug}}$ of size J×KN, by placing each individual matrix one below the other. Then, $X_{\mathrm{aug}}$ is submitted to bilinear decomposition according to the extended model formulation of MCR-ALS [52]:(6)$$X_{\mathrm{aug}}=C_{\mathrm{aug}}S^{T}+E_{\mathrm{aug}}$$

After convergence, S captures the profiles of *N* species in the non-augmented mode (generally, the spectral mode) which is common to all samples, whereas $C_{\mathrm{aug}}$ comprises the profiles of *N* species in the augmented mode, in each of the submatrices of $X_{\mathrm{aug}}$ (generally, the elution time mode). Additionally, the area under the profiles captured within $C_{\mathrm{aug}}$ is tied to the relative contributions of the individual components in each submatrix, which are then coupled to a regression model for quantitative purposes. 

The possibility of performing a bilinear decomposition of augmented data arrays constitutes the fundamental aspect that gives this model both the optimal flexibility and versatility to deal with multilinearity deviation problems that characterize chromatographic data. This fact is key to understand why MCR-ALS has originated a myriad of applications in analytical calibration and has been extended to different kind of data, including third-order data.

For the specific case of four-way calibration, an alternative strategy known as *A*PARAFAC was proposed by Bortolato et al. [13] combining the benefits of the PARAFAC uniqueness and the flexibility of MCR-ALS. This latter model can be implemented through a typical ALS process for performing a trilinear decomposition of an augmented three-way array. In this way, it allows to deal with non-quadrilinear type 1 data. The corresponding model for the ${\underline{X}}_{\mathrm{aug}}$ array of size *IJ* × *K* × *L* a can be represented by: (7)$${\underline{X}}_{\mathrm{aug}}=AB_{3W}{\left(D_{3W}⊙C_{3W}\right)}^{T}+{\underline{E}}_{\mathrm{aug}}$$
where the results of the decomposition are collected into three loading matrices **AB**_3W_ (*IJ* × *N*), **C**_3W_ (*K* × *N*) and **D**_3W_ (*L* × *N*). The model residuals are retained in the ${\underline{E}}_{\mathrm{aug}}$(*IJ* × *K* × *L*) array. Here, **AB**_3W_ collects the unfolded profiles along the augmented chromatographic elution time modes and brings the relative contribution of the individual constituent present in every sample. The remaining decomposition matrices contain the profiles that enable a qualitative interpretation; for example, for LC-EEM data **C** and **D** will contain the excitation and emission spectral profiles of each responsive component. 

Finally, the third category of multi-way chemometric approaches is constituted by latent variable-based models, essentially, the regression variants U-PLS and N-PLS. The PLS model was originally conceived for first-order calibration, i.e., senso strictum is only able to exploit the first-order advantage [53]. However, the PLS philosophy was extended to higher-order calibration, where the second-order advantage can be achieved by coupling the model to the RML methodology [54].

In U-PLS/RML, during the calibration step, a classical PLS model is built with a set of unfolded multi-way data, disposed in a matrix $X_{\mathrm{PLS}}$, which is then subjected to the following decomposition [53]: (8)$$X_{\mathrm{PLS}}=PT^{T}+E_{\mathrm{PLS}}$$

In Equation (8), P and T are the PLS loading and score matrices, respectively, which aim to maximize both the explained variance in the $X_{\mathrm{PLS}}$ data and the covariance with the nominal analyte concentration in the calibration samples. $E_{\mathrm{PLS}}$ captures the PLS model residuals [55]. The PLS model returns a vector of latent regression coefficients v, which is usually employed to make predictions in test samples according to:(9)$$y=v^{T}t$$
where t is the score vector of a given test sample, obtained by projection of an unfolded test signal onto the space spanned by the PLS calibration loadings. If a given test sample contains unmodelled constituents, the PLS prediction residual might be abnormally large compared to the expected noise level [53]. Under these circumstances, the RML methodology intends to decompose the part of the test data unexplained by PLS, assuming that the residuals can be rearranged into a multilinear array. For second-order calibration, RML is usually referred as RBL (bilinear) and its mathematical expression for a given test sample $X_{\mathrm{test}}$ can be formulated as [54]:(10)$$X_{\mathrm{test}}=\mathrm{reshape}\left(Pt_{\mathrm{RBL}}\right)+B_{\mathrm{RBL}}T_{\mathrm{RBL}}^{T}+E_{\mathrm{RBL}}$$
where $B_{\mathrm{RBL}}T_{\mathrm{RBL}}^{T}$ derives from the PCA model for the residual matrix $X_{\mathrm{test}}-\mathrm{reshape}\left(Pt_{\mathrm{RBL}}\right)$ with $n_{\mathrm{RBL}}$ principal components. During the RBL procedure, a new sample score vector $t_{\mathrm{RBL}}$ is calculated which only represents the analyte information. The $t_{\mathrm{RBL}}$ vector is then used to make analyte predictions through Equation (9). RTL is a natural extension of Equation (10) for third-order data. On the other hand, N-PLS/RML represents a multi-way variant of PLS and is based on an analogous fundamental to those of U-PLS/RML. The key difference is that the original multi-way data structure is preserved during the PLS calibration/prediction stages. 

From the qualitative point of view, in contrast to other model families, PLS loadings have no direct chemical interpretability. However, both latent models have shown to be advantageous in specific calibration scenarios. In particular, U-PLS/RML is the most flexible model and can be appropriate to model certain types of non-multilinear data where the lack of multilinearity occurs in more than one experimental mode. In principle, this is partially true if the second- or third-order advantage is to be achieved. Although PLS can satisfactory model a non-multilinear calibration dataset, RML procedures may fail during sample prediction if the residual matrix is not multilinear. In addition, for the particular case of chromatographic data, these methods are only applicable if the shifts in chromatographic profiles among runs are small. This means that U-PL/RML and N-PLS/RML are sensitive to the lack of multilinearity along the sample mode.

In order to graphically summarize all the information described in Section 2 and Section 3, the flow-chart shown in Figure 3 describes the most important modelling approaches for calibration purposes with chromatographic multi-way data, which have been here considered. If the generated data are multilinear, any of the presented models can in principle be implemented. However, multilinear models such as PARAFAC and AMLD variants should in this case be the first choice. When the data are not multilinear, or multilinearity cannot be restored by implementing convenient pre-processing procedures, MCR-ALS, PARAFAC2 or *A*PARAFAC (for four-way arrays) are the most appropriate models to be applied. In any case, minor multilinear deviations can be tackled by PLS models. In all cases, the concentration of one or more analytes can be simultaneously obtained, even in the presence of unexpected sample constituents. In the case of PLS models, the concentration of one or more analytes are directly calculated by means of RML procedures. On the other, in multilinear decomposition and curve-resolution models, the relative contribution of the calibrated analytes is represented by the so-called component scores. They can be coupled to a pseudo-univariate regression model to estimate the analyte concentration in unknown samples. 

## 4. Applications of LC Multi-Way Data

### 4.1. Second-Order/Three-Way Chromatographic Calibration

The continuous publication of scientific reports devoted to obtaining second-order/three-way chromatographic data and their valuable analytical applications makes it relevant to update the subject. As it is known, the analysis time demanded for both sample pre-treatment step and the chromatographic run itself is drastically reduced when multi-way calibration methods are implemented, while obtaining accurate quantitative information. Overcoming the problems inherent to chromatographic analysis of multi-component systems, such as co-elution of analytes and/or non-calibrated components and elution time shifts among runs, mostly depends on the proper selection of the chemometric model to be applied.

Table 1 summarizes some examples reported from 2018 to date developed for the analyte determinations based on second-order/three-way chromatographic calibration. The investigated analytes and matrices, the model/s selected for data processing, some relevant characteristics of the system and the attained limits of detection are indicated. As can be seen, the most widely applied approach for second-order/three-way chromatographic data generation is the popular LC-DAD. This may be due to a number of reasons, e.g., (1) most organic compounds absorb electromagnetic radiation in some region of the UV-visible spectrum, (2) modern DAD in chromatographic instruments enjoy a great sensitivity and versatility, and (3) it requires accessible equipment and low-cost consumables.

On the other hand, since not all compounds show fluorescent properties, it is not surprising that the number of publications describing chromatographic second-order calibration with fluorescence detection is much smaller. As will be seen below, in multivariate calibration, fluorescent detection has been more popular for obtaining third-order chromatographic data, where the EEM provides two instrumental modes while the third mode is given by the elution time. This detection mode takes advantage of the maximum selectivity and sensitivity which is gained by scanning both the emission and the excitation spectra, in comparison with fluorescence detection at a single excitation wavelength. In the latter case, a compromise is required among the excitation maxima of all analytes.

Finally, a smaller number of studies applying second-order LC data with MS detection have been published in the evaluated time period in comparison with DAD, despite the great potential of the former methodology. Most GC-MS second-order calibration publications shown in Table 1 involved quantitative studies of the migration of selected analytes from packaging material to foods, food simulants and cosmetic creams. LC-MS was applied to a variety of multi-components systems and complex samples.

Regardless of the detection method, the most used models for data processing were PARAFAC, ATLD and MCR-ALS. The advantages obtained when carrying out the chromatographic quantitation of analytes in complex samples using second-order data, as well as the problems that arise in each analysis, can be found in Table 1 for each report.

### 4.2. Third-Order/Four-Way Chromatographic Calibration

The quantitation of analytes through third-order/four-way data is not extensively applied by the analytical community. Although some works on this topic have recently been developed and their remarkable advantages as powerful analytical tools have been steadily highlighted, the number of bibliographic citations is still significantly lower than those corresponding to second-order calibration.

Applications of this type of high-order quantitation in the evaluate period are shown in Table 2. As can be appreciated, the most frequent way to generate third-order/four-way chromatographic data consists in measuring LC-EEM data.

While fluorescence matrices are easily obtained in fluorescent systems, their coupling to the chromatographic elution time mode must be carefully evaluated in order to generate a four-way array that complies with the properties required by the selected model for successful data processing [1,28,93,94,95]. The generation of photoinduced fluorescence in systems with analytes that do not present native fluorescence has been achieved through the incorporation of a UV reactor in the chromatographic equipment [93].

The research group led by Synovec has carried out an important contribution in the gas chromatography area for obtaining high-quality third-order data involving GC^2^-TOFMS. In recent years, the following works may be mentioned: (1) the study of the influence of the column selection and modulation period in the trilinear deviation ratio (TDR) range and, consequently, in the correct PARAFAC deconvolution [30]; (2) the use of a pulse flow valve for an ultra-fast modulation period and an innovative data processing method coupled with MCR-ALS deconvolution [96]; and (3) the advantages of partial modulation in the negative pulse mode [97]. In addition, valuable improvements to comprehensive GC^3^-FID detection have been achieved by the same research group [98,99].

## 5. Analytical Figures of Merit 

Ideally, every development of a new calibration protocol should be accompanied by a report providing the AFOMs. This is usually the case in univariate calibration, but unfortunately the practice is not universally extended to multi-way calibration procedures. The subject has been thoroughly reviewed in 2014 [14], but new developments have taken place in recent years, which deserve to be commented in the present review.

Since we advocate for the use of MCR-ALS as the model of choice for multi-way chromatographic data, the AFOMs will be discussed in the context of this chemometric tool [100]. Specifically, the component-wise classical parameters sensitivity (SEN*_n_*), analytical sensitivity (g*_n_*), selectivity (SEL*_n_*), limit of detection (LOD*_n_*) and limit of quantitation (LOQ*_n_*) will be discussed, together with a recently proposed figure of merit which is characteristic of MCR-ALS: the rotational ambiguity derived uncertainty (*RMSE*_RA_) [101].

The SEN*_n_* can be described in qualitative terms as the net analyte signal at unit concentration, which can be estimated for a specific sample component in MCR-ALS as [14]:(11)$${\mathrm{SEN}}_{n}=\frac{m_{n}}{J^{1/2}}{\left\{\;\delta_{n}^{T}\;{\left[\;S_{\exp}^{T}\left(\;I-S_{\mathrm{unx}}S_{\mathrm{unx}}^{+}\right)\;S_{\exp}\right]}^{-1}\delta_{n}\right\}}^{-1/2}$$
where *J* is the number of sensors of each sub-matrix in the augmented mode, *m_n_* is the slope of the pseudo-univariate calibration line, **S**_exp_ is the matrix of profiles in the non-augmented mode for expected constituents in the calibration set, **S**_unx_ is the matrix of profiles in the non-augmented mode for the unexpected constituents (interferents) and **d***_n_* is an analyte selector vector, having a value of 1 at the analyte index and 0s otherwise.

The Equation (10) is written in such a way that it resembles the SEN*_n_* definition in terms of net analyte signal. An equivalent expression can be written in a more compact manner as [100]:(12)$${\mathrm{SEN}}_{n}=\frac{m_{n}}{J^{1/2}}{\left(S^{T}S\right)}_{nn}^{-1/2}$$
where the subscript ‘*nn*’ indicates the (*n*,*n*) element of a matrix.

More useful than the plain SEN*_n_* is the analytical sensitivity *γ_n_*, defined as the ration between SEN*_n_* and the instrumental noise level *σ_x_* [102]:(13)$$\gamma_{n}=\frac{{\mathrm{SEN}}_{n}}{\sigma_{x}}$$
because it has inverse concentration units and is independent on the type of measured signal.

The SEL*_n_*, in turn, can be estimated as the ratio between the SEN*_n_* and the slope of the analyte calibration graph:(14)$${\mathrm{SEL}}_{n}=J^{1/2}\frac{{\mathrm{SEN}}_{n}}{m_{n}}$$

The degree by which the product (*J*^1/2^ SEN*_n_*) departs from *m_n_* in the latter equation depends on the level of overlapping among the component profiles. The value of SEL*_n_* varies between 0 (null selectivity) and 1 (100% or full selectivity). In the chromatographic context, multi-way selectivity has shown to be directly related to the effective peak capacity of a chromatogram [103].

The definition of the LOD*_n_* has been changing with time [104]. The old concept of LOD*_n_* as three times the standard deviation for a blank sample has been abandoned by IUPAC, in favour of the modern view of the LOD*_n_* as a function of Type I and Type II errors (also called a and b errors, or false positive and false negative errors) [105,106]. In addition, today it is widely accepted that the uncertainties in the measurement of the test sample should be added to those stemming from the uncertainties affecting the calibration procedure (including both the measurement of the instrumental signal and the preparation of the calibration standards).

Since MCR-ALS quantitation is usually performed through a pseudo-univariate procedure in which the test sample score is interpolated in a regression line of calibration scores against nominal analyte concentrations, the LOD*_n_* can be estimated from the latter line by extending the procedure, which is usual in univariate calibration, i.e.,
(15)$${\mathrm{LOD}}_{n}=3{.3(\mathrm{SEN}}_{n}^{-2}\sigma_{x}^{2}+h_{0}{\;\mathrm{SEN}}_{n}^{-2}\sigma_{x}^{2}+h_{0}\sigma_{y\mathrm{cal}}^{2}{)}^{1/2}$$
where *h*_0_ is the leverage for the blank sample, and the factor 3.3 is equal to (*t*_a,n_ + *t*_b,n_) for a = 0.05 and b = 0.05 and a large value of *n* (a and b are the probabilities for Type I and Type II errors, respectively and *n* is the number of degrees of freedom), *s_x_* is a measure of the uncertainty in the instrumental signal, which is considered to stem from identically and independently distributed noise in the signal measurements, and *s_y_*_cal_ is the concentration uncertainty when preparing the calibration samples. The factor 3.3 in the latter equation may be changed, if needed, for other values of a and b. Notice the that the assumptions underlying the expression for the limit of detection are: (1) the LOD*_n_* is close enough to the blank so that the leverage can be considered equal to *h*_0_, and (2) the distance from the blank to the LOD*_n_* is a sum of two confidence intervals; a more rigorous treatment suggests the use of a non-centrality parameter of a non-central *t* distribution instead of a sum of classical *t*-coefficients [107]. 

Under similar assumptions, the LOD*_n_* is defined as the concentration for which the relative prediction uncertainty is 10%, leading to:(16)$${\mathrm{LOQ}}_{n}{=10(\mathrm{SEN}}_{n}^{-2}\sigma_{x}^{2}+h_{0}{\;\mathrm{SEN}}_{n}^{-2}\sigma_{x}^{2}+h_{0}\sigma_{y\mathrm{cal}}^{2}{)}^{1/2}$$

Specifically, for MCR-ALS, a recently proposed figure of merit is required for analytical reports based on the use of this model: the rotational ambiguity uncertainty. The latter stems from the fact that all bilinear decomposition solutions, such as those provided by MCR-ALS, are not necessarily unique, even when all possible constraints are applied during the decomposition [41]. When this is the case, i.e., for non-unique solutions, an uncertainty remains in the estimation of the analyte concentration, because a range of feasible solutions are possible [108]. The *RMSE*_RA_ derived from the existence of the feasible solutions has been shown to be given by the following range [101]:(17)$$RMSE_{\mathrm{RA}}=\left[\frac{\delta_{\mathrm{RA}}}{\sqrt{12}};\frac{\delta_{\mathrm{RA}}}{\sqrt{3}}\right]$$
where *δ*_RA_ is the width of the range of feasible concentration values for the analyte, which, in turn, is equal to the difference between maximum and minimum areas of the analyte concentration profile, converted to concentration units through the calibration slope:(18)$$\delta_{\mathrm{RA}}=\frac{max\left(a_{\mathrm{test}}\right)-min\left(a_{\mathrm{test}}\right)}{m_{n}}$$

It is important to notice that, even when a significant *RMSE*_RA_ value can be computed by the latter equation, the MCR-ALS solutions may be driven to the correct bilinear solution by an adequate initialization procedure, as recently discussed [109]. In these cases, the value of *RMSE*_RA_ may be large, although this would not be reflected in the average prediction error for a set of test samples, which may ultimately be reasonably low. However, it is important to notice that for future test samples, having a varying chemical composition in terms of interferents, the chosen initialization may not lead to similarly good results. The subject has been explained in detail in a recent tutorial [41].

Finally, if PARAFAC or other AMLD models are employed to process multi-way data of chromatographic origin, the decomposition is often unique, with AFOMs that have already been discussed in detail [14,110]. These models may successfully process data where the position and shapes of the chromatographic peaks do not change across samples, but as explained above it cannot be considered as the tool of choice for the presently discussed multi-way data.

## 6. An Example Comparing Second- and Third-Order Data

It is known, from theoretical considerations, that the SEN*_n_* and SEL*_n_* should increase with the increasing number of data modes when decomposing a multi-way array [14]. However, only in a few cases, this has been experimentally confirmed, by registering, for the same analytical system, second- and third-order data, and then comparing the resulting figures of merit [25].

Recently, such a comparison was made in connection with the simultaneous quantitation of various analytes in complex food samples [94]. The applied protocols were based on LC with fluorescence detection, employing two different detection setups for the same chromatograph. In one of them, second-order data were measured with fluorescence emission detection at a fixed excitation wavelength. In the second one, third-order data were collected by EEM detection. The analytes were ten different quinolone antibiotics, which were simultaneously analysed in edible animal tissues (chicken liver, bovine liver and bovine kidney). MCR-ALS was applied to augmented matrices for each test sample along the elution time direction. The use of MCR-ALS as data processing model provided excellent results with second-order data; the relative prediction errors spanned the range 4–12% in the case of the real samples at analyte concentrations, which are compatible with the maximum residue levels accepted by international regulations. 

Third-order data were expected to provide better results, on account of the known sensitivity and selectivity increase by increasing the data order. In this case, the raw three-way arrays for each sample were unfolded into a matrix, by concatenating the emission and excitation modes into a single one prior to building the augmented data matrix along the elution time direction. The authors noticed that the third-order MCR-ALS analytical results were, in general, worse than for second-order data, with relative errors in the range 9–23%, and significantly larger detection limits for some analytes (Table 3). Furthermore, one of the analytes could not be resolved. Even when the theoretically estimated SEN*_n_* was larger for third-order data (Table 3), as expected from the increase in the number of data modes, the relevant statistical indicators for analyte estimation (average prediction error and relative error of prediction) and the detection capabilities were indeed worse (Table 3). 

Subsequently, the latent structured models U-PLS/RBL and U-PLS/RTL were applied to the second- and third-order data, respectively [94]. In this case, relative errors in the range 7–18% were obtained for second-order calibration and 5–27% for third-order calibration. The latter errors were significantly larger than those for MCR-ALS/second-order data, although the U-PLS/RTL model permitted the detection of all the studied analytes.

It was concluded from the comparison of these two different experimental methodologies, carried out by using different detectors, that matrix fluorescence detection in LC would still require some advances in instrumental setup, in order to provide maximum sensitivity and signal/noise ratio at very high scanning rates, something which is not currently possible with commercially available equipment. In the above commented report, the second-order data were measured using a commercial instrument which is fully optimized for maximum sensitivity, whereas third-order data collection required the in-house connection of a LC to a conventional spectrofluorometer, which is not optimized for emission measurements in a small flowing cell.

This example also illustrates the fact that the plain sensitivity parameter SEN*_n_* may not be used for a proper comparison of analytical methods based on data measured with different instruments. As explained above, a better AFOM is the *γ_n_*. The effect of a larger noise level could be the ultimate reason why third-order calibration failed to provide better analytical results in the example of ref. [94].

## 7. Conclusions

Chromatography is an ever-growing discipline. Advances in experimental activities have demanded the development of new chemometric models, whose ultimate aim is to extract useful information for developing powerful analytical protocols. This is particularly true in the area of multi-way chromatography, which is able to produce large data sets that can be arranged into mathematical objects of increasing complexity. Selecting the appropriate multi-way chemometric model to process these data sets is crucial for achieving one of the most important advantages offered by this field, i.e., the possibility of quantitating analytes in complex samples containing uncalibrated interferents (the second-order advantage). The present review provides a current view of the state-of-the-art, both from experimental and theoretical perspectives.

## Figures and Tables

**Figure 1 molecules-26-06357-f001:**
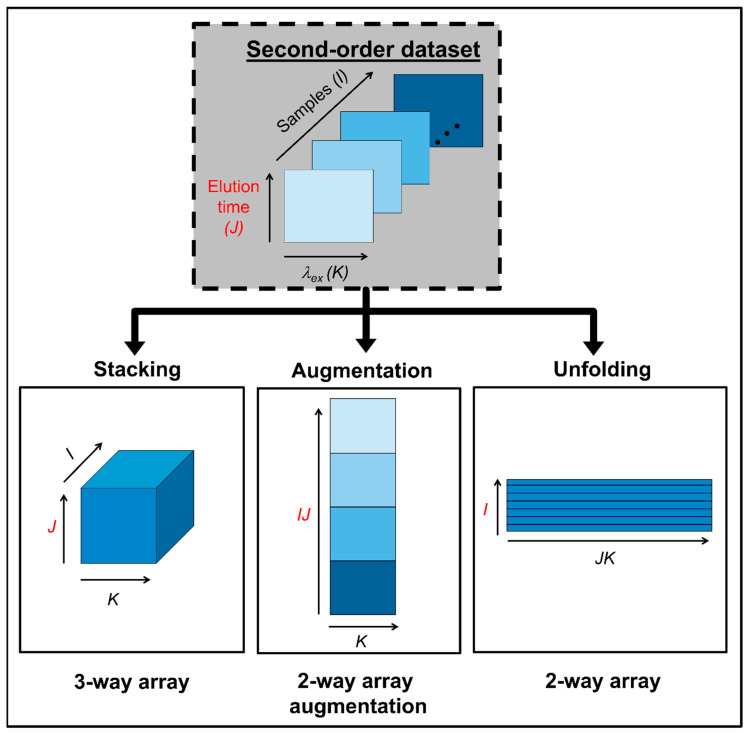
Possible data arrays that can be built with second-order data prior to chemometric modelling.

**Figure 2 molecules-26-06357-f002:**
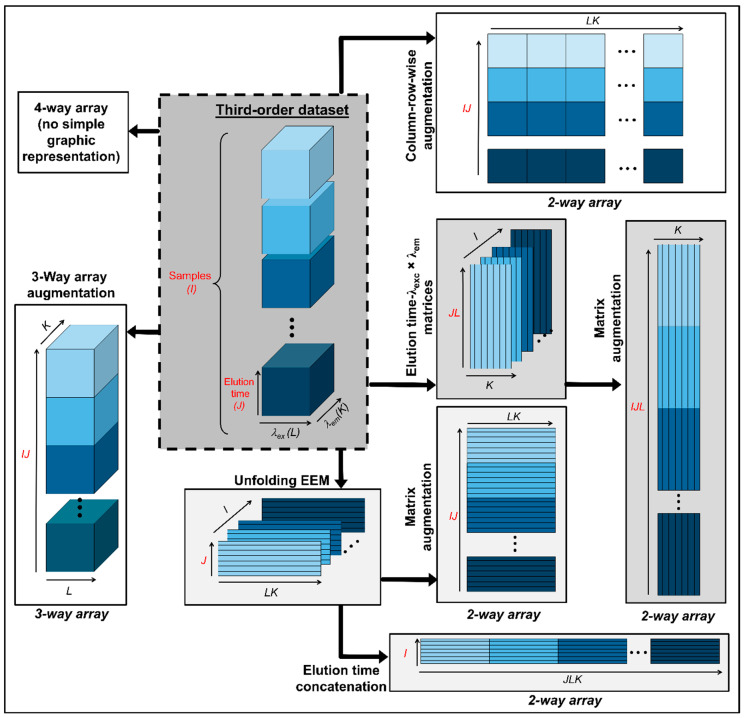
Possible data arrays that can be built with third-order data prior to chemometric modelling.

**Figure 3 molecules-26-06357-f003:**
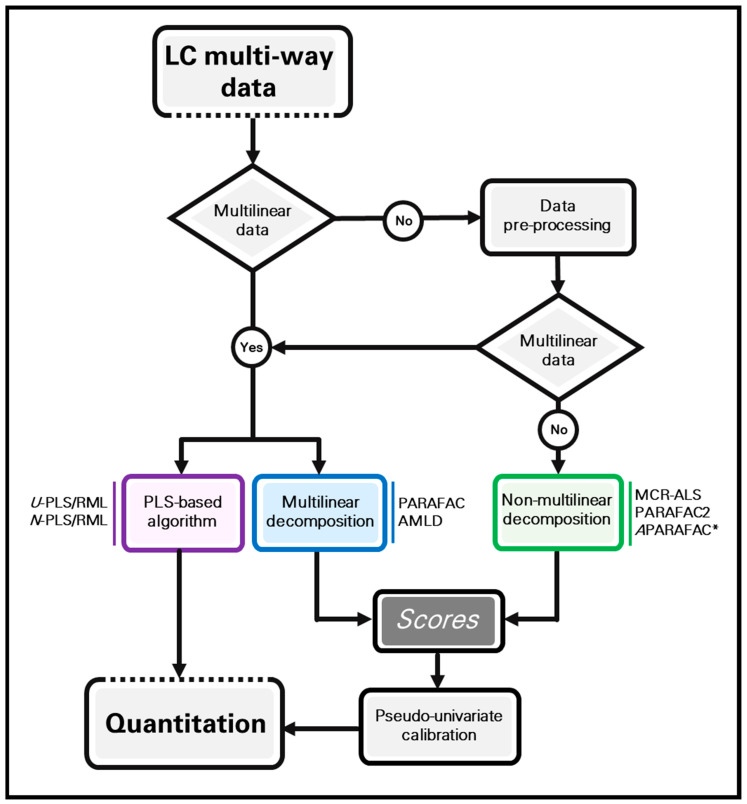
Flow-chart showing different modelling approaches to analyse LC multi-way data with quantitative aims. All the models are indicated as examples. *A*PARAFAC* is implemented only for third-order data analysis.

**Table 1 molecules-26-06357-t001:** Reports from 2018 to date based on chromatographic second-order/three-way data for quantitation purposes.

Analytes and Samples	Model	Remarks	Ref.
**LC-DAD**			
Malic, oxalic, formic, lactic, acetic, citric, pyruvic, succinic, tartaric, propionic and α-cetoglutaric acids in yoghurt, cultured milk, cheese and wine	PARAFAC U-PLS/RBL	ET 10 min (isocratic mode). LODs: 0.15–10.0 mmol L^−1^ in validation samples	[56]
Meloxicam, flurbiprofen, phenylbutazone, ibuprofen, diclofenac, mefenamic acid, celecoxib, naproxen, ketoprofen, diflunisal (non-steroidal anti-inflammatories) in Chinese patent drugs and health products	ATLD	ET 14.5 min (gradient elution). To simplify data processing, the retention time mode was subdivided in four regions. LODs: 0.01–0.12 μg mL^−1^	[57]
Chlorogenic acid, (−)-epicatechin, caffeic acid, taxifolin, p-coumaric acid, hesperetin, naringenin, chrysin, apigenin, kaempferol, luteolin, quercetin, myricetin, rutin, (+)-catechin, ferulic acid, isorhamnetin (polyphenols) in raw propolis	ATLD	ET 16.5 min (gradient elution). To simplify data processing, the elution time mode was subdivided in eight regions. LODs: 0.01–0.38 μg mL^−1^	[58]
Gliclazide, glibenclamide, glimepiride (antidiabetics), atenolol, enalapril, amlodipine (antihypertensives) in serum	MCR-ALS U-PLS/RBL	ET 3 min (isocratic mode). Elution time mode was subdivided in two regions. LODs: <30 ng mL^−1^; better for U-PLS/RBL	[59]
Tacrolimus, everolimus, cyclosporine A (immunosuppressants) in whole blood	MCR-ALS	ET 2.7 min (isocratic mode). Minimum sample preparation steps. The time mode was subdivided in three regions. Sample-added calibration strategy for matrix effect. LODs: 0.56 μg L^−1^ (tracolimus), 0.08 µg L^−1^ (everolimus), 7.6 µg L^−1^ (cyclosporine A)	[60]
Methylparaben, ethylparaben, propyl-paraben, butylparaben, phenoxyethanol salicylic acid, methylisothiazolinone, 3-iodo-2-propynyl-n-butylcarbamate (preservatives) in facial masks	ATLD MCR-ALS	ET 8.2 min (gradient elution). Elution time mode was subdivided in four regions. Satisfactory and statistically comparable results were obtained with both models, except for phenoxyethanol in one studied sample (this fact was attributed to matrix interferences). LODs: 1.2 ng mL^−1^ (butylparaben), 1466 ng mL^−1^ (3-iodo-2-propynyl-*n*-butylcarbamate)	[61]
Prednisolone, methylprednisolone (corticosteroids), mycophenolic acid (immunosuppressant) in human plasma	MCR-ALS	ET < 4 min. Two isocratic elution methods with two different mobile phases were used. Data array was divided into two regions (matrix augmentation in spectra and retention time direction were implemented for the first and second regions, respectively). LODs: 0.9 µg L^−1^ (prednisolone), 1.3 µg L^−1^ (methylprednisolone), 300 µg L^−1^ (mycophenolic acid)	[62]
1,2-Dinitrobenzene, 1,3-dinitrobenzene, 2,4,6-trinitrotoluene, 2,4-dinitrotoluene, 2-nitrotoluene, 3-nitrotoluene and 4-nitrotoluene (explosives, agrochemical, textile dyes and chemical intermediates) in river and pond waters	MCR-ALS	ET 10 min (isocratic mode). Very similar analyte structures. LODs: 0.05–0.12 µg mL^−1^	[63]
Uric acid, creatinine, tyrosine, homovanillic acid, hippuric acid, indole-3-acetic acid, tryptophan, 2-methylhippuric acid (small molecules related to early diseases diagnosis) in human urine	ATLD MCR-ALS	ET 6 min (isocratic mode). Both models rendered comparable recoveries and root mean square error of predictions. LODs: 29.9–464.2 ng mL^−1^ (ATLD); 11.7–127.1 ng mL^−1^ (MCR-ALS)	[64]
Chrysene, naphtalene, acenaphthylene, fluorene, phenanthrene, acenaphthene, anthracene, pyrene, benzo[*a*]anthracene, guaiazulene, benzo[*e*]pyrene, fluoranthene, benzo[*a*]pyrene, benzo[*b*]fluoranthene, benzo[*k*]fluoranthene (PAHs) in flue-dust and greasy dirt samples	ATLD	ET 18 min (isocratic mode). InertSustain^®^-C18 (5.0 μm, 4.6 mm × 250 mm) reversed phase column. Elution region was divided into four sub-segments. LODs: 0.94–48.86 ng mL^−1^	[65]
Naproxen, ketoprofen, meloxicam (non-steroidal anti-inflammatories) in Chinese patent drugs	ATLD-MCR	ATLD-MCR model was compared with PARAFAC, ATLD and MCR-ALS. Two simulated HPLC-DAD data sets, one simulated LC-MS data set, and a semi-simulated LC-MS data set were evaluated. ATLD-MCR proved to be able for handling chromatographic data with time shifts and signal overlapping. ^b^	[66]
2-Hydroxy-4-methoxybenzophenone (UV filter) in mice serum and human plasma	MCR-ALS	ET 2 min (isocratic mode). Ultra-HPLC using two different experimental methods was applied. One of these methods rendered better results. LOD: 0.66 ng mL^−1^	[67]
Sixteen PAHs (US EPA priority pollutants) in river water	MCR-ALS	USAEME as extraction method. Region-based analysis had improvement regarding to the whole data analysis. LODs: 4.77–16.44 ng mL^−1^	[68]
Dorzolamide hydrochloride (carbonic anhydrase-II inhibitor) and timolol maleate (non-specific adrenergic blocker) in an ophthalmic solution	PARAFAC 3W-PLS U-PLS	ET 0.5 min (isocratic mode). LODs: 0.57 µg mL^−1^ (dorzolamide hydrochloride), 0.66 µg mL^−1^ (timolol maleate)	[69]
Tartrazine, sunset yellow, carmine, amaranth, brilliant blue, aspartame, acesulfame potassium, sodium saccharin, caffeine, benzoic acid, sorbic acid, glycyrrhizin acid (food additives) in beverages (cola, grape, lemon, and orange sodas, green and black teas, orange and apple juices, milk drinks and grape wine)	ATLD	ET 8.1 min (gradient elution). ET mode was subdivided in four regions. LODs: 1.40–165.1 ng mL^−1^	[70]
Gallic acid, epigallocatechin, epicatechin, epigallocatechin gallate, epicatechin gallate (polyphenols) in red, green, black and clinacanthus nutans Chinese teas	ATLD MCR-ALS ATLD-MCR	ET 5.4 min (isocratic mode). Data array was divided in two sub-regions. MCR-ALS and ATLD-MCR were better than ATLD in the case of larger time shifts	[71]
Carbendazim, thiabendazole, fuberidazole, carbofuran, carbaryl, flutriafol, 1-naphthol (pesticides) in vegetables (lettuce, cabbage leaf, carrot, beet, tomato, green bell pepper)	MCR-ALS	ET 25 min (gradient elution). Standard addition method due to matrix effect. Advantages of multi-way calibration in comparison with the univariate one when interferents are present (vegetable samples) were demonstrated	[72]
Epicatechin, myricetin, fisetin, quercetin, hesperidin, kaempferol, rutin (flavonoids) in raw and purified Chinese propolis	ATLD	ET 3.5 min (isocratic mode). LODs: 0.01–0.20 μg mL^−1^	[73]
Melamine (plastic) in a food simulant migrated from kitchenware	PARAFAC PARAFAC2	ET 2.2 min (isocratic mode). Both models avoided the overestimation of migrated melamine amount despite coelution of the interferent with the analyte. LOD: 0.58 mg L^−1^	[74]
Bisphenol A, bis(4-hydroxyphenyl) methane, 4,4′-cyclohexylidenebisphenol, 4,4′-(hexafluoroisopropylidene)diphenol, bis(4-hydroxyphenyl) sulfone [bisphenols] in methanol synthetic solutions	U-PLS	ET 4 min (isocratic mode). After quantitation, optimization of experimental conditions was made by inversion of PLS. LODs: 334–1156 μg L^−1^	[75]
4,4′-isopropylidenediphenol, 4,4′-methylenediphenol, 4,4′-cyclohexylidene bisphenol, 4,4′-(hexafluoroisopropylidene) diphenol, 4,4′-sulfonyldiphenol [bisphenols] in a food simulant	PARAFAC	ET 4 min (isocratic mode). Migration from BPA-free PC glasses is studied. The maximum amount of BPA migrated from PC glasses (5.60 μg L^−1^) was lower than the established migration limit for a non-authorized substance. LODs: 44.0–138.9 μg L^−1^.	[76]
Benzoic acid, methylisothiazolinone, sorbic acid, phenoxyethanol, methylparaben, ethylparaben, propylparaben, butylparaben, 3-iodo-2-propynyl-n-butylcarbamate, clorophene, triclocarban, triclosan (preservatives) in emulsion, cream, powder, gel and facial masks	ATLD	ET 10.5 min (gradient elution). LC-DAD data were divided into five temporal regions for data processing. LODs: 9.57 × 10^−4^–0.33 μg mL^−1^	[77]
Lenalidomide, gefitinib, crizotinib, chidamide, dasatinib, axitinib, lapatinib, erlotinib, nilotinib, idelalisib (anti-tumor drugs) in plasma, urine, cell culture medium	ATLD, MCR-ALS ATLD-MCR	ET 6.5 min (gradient elution). MCR-ALS and ATLD-MCR rendered better recoveries than ATLD. LODs: 5.4–398 ng mL^−1^ (ATLD-MCR), 0.1–536 ng mL^−1^ (MCR-ALS)	[78]
Imidacloprid, albendazole, fenbendazole, praziquantel, fipronil, permethrin (veterinary active ingredients) in water from a wetland system used for the treatment of waste from a dog breeding plant	MCR-ALS	ET 12 min (gradient elution). Optimized DLLME. Data array was divided into six regions. LODs: 1.3–8.5 ng mL^−1^	[79]
Gefitinib, crizotinib, dasatinib, axitinib, lapatinib, erlotinib, pexidartinib, nilotinib, LDK378 (tyrosine kinase inhibitors) in plasma, urine and a cell culture medium	ATLD-MCR	ET 7 min (gradient elution). Data were divided into three regions. LODs: 0.02–0.24 μg mL^−1^ (plasma), 0.01–0.14 μg mL^−1^ (urine), 0.02–2.44 μg mL^−1^ (cell culture)	[80]
Sudan I, sudan II, sudan III, sudan IV, sudan red B, sudan red G, sudan red 7B, para red, diethyl yellow, methyl red, butter yellow, toluidine red (edible azo dyes) in chili sauces, saffron, ketchup, chili powder	ATLD MCR-ALS	ET 6.5 min (isocratic mode). Three-way data array was divided into four regions on the basis of elution time. LODs: 0.01–2.56 mg kg^−1^ (ATLD), 0.01–2.95 mg kg^−1^ (MCR-ALS).	[80]
**LC-FLD**			
Pipemidic acid, marbofloxacin, enoxacin, ofloxacin, norfloxacin, ciprofloxacin, lomefloxacin, danofloxacin, enrofloxacin, sarafloxacin (quinolone antibiotics) in chicken liver, bovine liver and kidney	MCR-ALS	ET 4.7 min (isocratic mode). LC-FLD data were divided into three temporal regions for data processing. LODs: 7–125 μg kg^−1^	[81]
**GC-MS**			
Butylated hydroxytoluene (BHT) [antioxidant], diisobutyl phthalate (DiBP), bis(2-ethylhexyl) adipate (DEHA), diisononyl phthalate (DiNP) [plasticizers], benzophenone (BP) [UV stabilizer] in Tenax ^a^	PARAFAC	ET 19.1 min. Data were acquired in SIM mode using five acquisition windows. LODs 2.28 µg L^−1^ (BHT), 7.87 µg L^−1^ (DiBP), 3.04 µg L^−1^ (DEHA), 124.8 µg L^−1^ (DiNP), 10.57 µg L^−1^ (BP). Tenax could not be reused in this multiresidue determination	[82]
Butylated hydroxytoluene (BHT), benzophenone (BP), benzophenone-3 (BP3), diisobutyl phthalate (DiBP) [filters and additives] in sunscreen cosmetic creams	PARAFAC PARAFAC2	ET 15.1 min. Data were acquired in SIM mode using four acquisition windows. LODs 7.93 µg L^−1^ (BHT), 12.40 µg L^−1^ (BP), 11.65 µg L^−1^ (DiBP), 279.8 µg L^−1^ (BP3). Analyte identification using the univariate standard method was incorrect. Multivariate calibration avoided false negative results	[83]
Butylated hydroxytoluene (BHT) [antioxidant], diisobutyl phthalate (DiBP), bis(2-ethylhexyl) adipate (DEHA), diisononyl phthalate (DiNP) [plasticizers] and benzophenone (BP) [UV stabilizer] in Tenax ^a^	PARAFAC PARAFAC2	ET 19.1 min. Migration from PE, PVC, and PP is studied. Data were acquired in SIM mode. Five acquisition windows were considered. Presence of BHT, DiBP and DEHA was confirmed in Tenax blanks in some of the analysis. BP, DiBP migrated from both PVC film and PP coffee capsules, whereas DEHA migrated from PVC film LODs: 3.48–360.2 µg L^−1^.	[84]
Bisphenol A (plasticizer) in a food simulant migrated from polycarbonate tableware, dichlobenil (pesticide) in onion, and oxybenzone (aromatic ketone) in sunscreen cosmetic creams	PARAFAC PARAFAC2	Both models allowed analyte quantitation. The analyzed cases were: presence of interferents with overlapping peaks to the IS and analyte, coeluting compounds which share ions with the IS, retention time shifts from sample to sample, and coelution of interferents	[74]
Butylated hydroxytoluene (BHT) [antioxidant], diisobutyl phthalate (DiBP) [plasticizer], benzophenone (BP) [UV stabilizer] in coffee	PARAFAC	ET 19.1 min. Migration from plastic capsules is studied. SBSE for analyte extraction and concentration. Standard addition method due to matrix effect. Data acquired in SIM mode using three acquisition windows. Traces of the analytes found in the Milli-Q water samples were taken into account in the analysis. Found levels in coffee were below or around (DiBP case) than the migration established limits	[85]
Fluoranthene, benzo[*b*]fluoranthene, chrysene, benzo[*a*]anthracene, pyrene (PAHs) in aerosol samples collected from Loudi City (China) in functional zones	ATLD-MCR MCR-ALS	ET < 8 min. Filters sample were extracted with the Soxhlet method. Scan mode was used for mass spectrum detection. In real samples, ATLD-MCR provided results which were better than or similar to MCR-ALS. LODs: 0.003–0.087 µg mL^−1^.	[86]
**LC-MS**			
Betamethasone, dexamethasone, triamcinolone acetonide, cortisone 21-acetate, dexamethasone 21-acetate, budesonide, triamcinolone acetonide acetate, fluocinonide, clobetasol 17-propionate, betamethasone dipropionate, beclomethasone dipropionate, beclomethasone, fluoromethalone, fluticasone propionate, betamethasone 17-valerate (glucocorticoids) in face masks	ATLD	ET 11 min (gradient elution). ESI interface operating in positive mode. LC–MS analysis in full scan mode. The three-way data array was divided into six sub-regions. Betamethasone and dexamethasone (epimers) were simultaneously quantified under a simple elution program. LODs: 0.56–13.55 ng mL^−1^.	[87]
Thiamine, riboflavin, nicotinic acid, biotin, nicotinamide, D-pantothenic acid, pyridoxine, folic acid, cyanocobalamin (B-group vitamins) in energy drinks	ATLD APTLD	ET < 4.5 min (gradient elution). ESI interface operating in positive mode. LC–MS analysis in full scan mode. Data array was divided into three sub-regions. Both models rendered similar recovery and statistical results LODs: 2 × 10^−3^–2.5 × 10^−2^ µg mL^−1^ (ATLD), 1 × 10^−3^–2.5 × 10^−2^ µg mL^−1^ (APTLD).	[88]
Estriol, 17α-estradiol, 17β-estradiol, estrone, ethinyl estradiol, diethylstilbestrol (estrogens), bisphenol A (xenoestrogen) in infant milk powder	ATLD	ET < 7 min (gradient elution). ESI interface operating in negative mode. LC–MS analysis in full scan mode. Data array was subdivided into four sub-regions on the basis of the elution ranges of estrogen. LODs: 0.07–2.49 ng mL^−1^	[89]
Gallic acid, chlorogenic acid, caffeic acid, (+)-catechin, *p*-coumaric acid, taxifolin, (−)-epicatechin, ferulic acid, myricetin, luteolin, quercetin (polyphenols) in Chinese propolis	ATLD MCR-ALS	ET < 7.0 min (gradient elution). ESI interface operating in negative mode. LC–MS analysis in full scan mode. Data array was subdivided into six sub-regions on the basis of the retention time. LODs: 2.8–80.0 ng mL^−1^ (ATLD), 0.9–54.5 ng mL^−1^ (MCR-ALS)	[90]
Cyclosporine-A and tacrolimus (immunosuppressants) in blood and surface water	MCR-ALS	ESI interface operating in positive mode. LC–MS analysis in full scan mode. The regions of interest method of the LC–MS data was employed for data compression. Matrix-matched calibration strategy was employed due to matrix effect. LODs: (blood) 5.8 ng mL^−1^ (cyclosporine-A), 4.8 ng mL^−1^ (tacrolimus). LODs (water) 2.3 × 10^−2^ ng mL^−1^ (cyclosporine-A), 9.0 × 10^−2^ ng mL^−1^ (tacrolimus)	[91]
17-β-estradiol, estrone, diethylstilbestrol (estrogens), bisphenol A (xenoestrogen) in river water [system I], L-glutamic acid, L-tyrosine, L-tryptophan, L-phenylalanine (amino acids), xanthine, hypoxanthine (purines), kynurenic acid, L-kynurenine (metabolites) in human urine [system II]	ATLD	Combination and partition of the MS1 full scan ion peaks recorded at different fragmentor voltages. Combined data and partitioned in two ways were compared using two systems. *System I:* ET 4.4 min (gradient elution) LODs: 0.18–2.72 ng mL^−1^(combined data), 0.25–2.35 ng mL^−1^ (partitioned data, ofv), 0.04–0.54 ng mL^−1^ (partitioned data, hfv). *System II*: ET 4.4 min (gradient elution). LODs: 0.99–5.43 ng mL^−1^ (combined data), 0.04–7.69 ng mL^−1^ (partitioned data, ofv), 2.96–7.38 ng mL^−1^ (partitioned data, hfv). In most cases, data combination rendered higher sensitivity and more reliable results. Data partition provided higher selectivity in some cases but in others was unable to quantify analytes	[92]

^a^ Tenax: food simulant for testing migration from plastics into dry foodstuffs. ^b^ For sample pre-processing and elution conditions in real samples see ref. [66]. Abbreviations: APTLD, alternating penalty trilinear decomposition; DLLME, dispersive liquid-liquid microextraction; ESI, electrospray ionization; ET, elution time; hfv, highest fragmentor voltage; IS, internal standard; LOD, limit of detection; ofv, optimum fragmentor voltage; PAH, polycyclic aromatic hydrocarbon, PE, polyethylene; PP, polypropylene; PVC, polyvinyl chloride; SBSE, stir bar sorptive extraction; SIM, single ion monitoring; USAEME, ultrasonic assisted emulsification micro-extraction; US EPA, United States Environmental Protection Agency; 3W-PLS, three-way partial least-squares.

**Table 2 molecules-26-06357-t002:** Reports from 2018 to date based on chromatographic third-order/four-way data for quantitation purposes.

Analytes and Samples	Model	Remarks	Ref.
**LC-EEFM**			
Rimsulfuron (herbicide), fuberidazole (fungicide), carbaryl (insecticide), naproxen (non-steroidal anti-inflammatory), albendazole (antihelminthic agent), tamoxifen (anticancer agent) in well and river waters	MCR-ALS	ET < 12 min (gradient elution mode). Native and photoinduced fluorescence (using a post-column UV reactor) were measured. Quadrilinearity was broken due to temporal shifts. LODs: 0.02–0.27 ng mL^−1^ (spiked water samples) after a preconcentration SPE step	[93]
Benz[a]anthracene, chrysene, benzo[b]fluoranthene, benzo[a]pyrene (PAHs) in tea leaves	MCR-ALS	ET ~ 9 min (isocratic mode). Non-quadrilinear LC-EEM data type 4 were successfully processed with an MCR-ALS strategy. LODs: 1.0–1.4 ng mL^−1^ (validation samples). LODs: 1.3–2.9 ng mL^−1^ (samples with interferents)	[28]
Pipemidic acid, marbofloxacin, enoxacin, ofloxacin, norfloxacin, ciprofloxacin, lomefloxacin, danofloxacin, enrofloxacin, sarafloxacin (quinolone antibiotics) in animal tissues (chicken liver, bovine liver and kidney)	MCR-ALS U-PLS/RTL	Third-order/four-way data results were compared with second-order/three-way data for the same system employing two different fluorescence detectors. MCR-ALS gave suitable results with second-order data but could not resolve all the analytes in the third-order/four-way system. U-PLS (with RBL or RTL) rendered good results in both cases, but the statistical indicators were not better than MCR-ALS second-order data. For more discussion, see section on Figures of Merit	[94]
Pyridoxine (vitamin B6) in the presence of L-tyrosine, L-tryptophan, 4-aminophenol in synthetic aqueous samples	PARAFAC APARAFAC	ET ~ 200 sec (isocratic mode). Multilinearity was restored by chemometric processing. LOD: 7 mg L^−1^	[95]
**GC^3^/univariate detection**			
Citronellol, eugenol, farnesol, geraniol, menthol, trans-anethole, carvone, β-pinene (allergens) in perfumes	PARAFAC	Two detectors (FID and a mass analyzer) were used for data acquisition. GC^3^ system involved a first modulator (thermal desorption modulator, mp 6 s) that interfaced the first two columns, and a second modulator (differential flow modulator, mp 300 ms) which connected the last two columns. For data processing, smaller subsections of the chromatogram were used. LODs (GC^3^-FID): 2.1–6.8 μL L^−1^, LODs (GC^3^-MS): 4.8–8.5 μL L^−1^	[31]

Abbreviations: GC^3^, three-dimensional gas chromatography, ET, elution time; FID, flame ionization detector; LOD, limit of detection; mp, modulation period; PAH, polycyclic aromatic hydrocarbon; SPE solid phase extraction.

**Table 3 molecules-26-06357-t003:** AFOMs from second- and third-order calibration of a number of fluoroquinolone antibiotics using MCR-ALS. Data from ref. [94].

Analyte ^a^	Second-Order			Third-Order		
	SEN ^b^	LOD ^c^	LOQ ^c^	SEN ^b^	LOD ^c^	LOQ ^c^
PIPE	145	0.01	0.03	150	0.1	0.3
MARBO	37	0.07	0.2	450	- ^d^	- ^d^
ENO	9	0.1	0.3	N.F. ^e^	N.F. ^e^	N.F. ^e^
OFLO	180	0.04	0.1	2100	0.04	0.1
NOR	226	0.01	0.03	340	- ^d^	- ^d^
CIPRO	26	0.03	0.1	86	- ^d^	- ^d^
LOME	48	0.03	0.1	650	0.1	0.3
DANO	230	0.01	0.03	4000	0.01	0.03
ENRO	38	0.01	0.03	720	0.08	0.2
SARA	120	0.02	0.06	910	0.1	0.3

^a^ PIPE, pipemidic acid, (MARBO), marbofloxacin, (ENO), enoxacin, (OFLO), ofloxacin, (NOR), norfloxacin, (CIPRO), ciprofloxacin, (LOME), lomefloxacin, (DANO), danofloxacin (ENRO), enrofloxacin, (SARA), sarafloxacin. ^b^ SEN = sensitivity in arbitrary fluorescence units × L mg^–1^. ^c^ LOD = limit of detection, LOQ = limit of quantitation, both in mg L^–1^. ^d^ LOD values are close to the maximum calibration concentrations. ^e^ N.F. = not found.

## Data Availability

Not applicable.

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
