# Peer review of "Chromatographic Applications in the Multi-Way Calibration Field"

_molecules, 2021, doi:10.3390/molecules26216357_

Round 1
Reviewer 1 Report
Please see the attachment.

Author Response
Comment
The research group has done a lot of excellent work in the field of multi-way calibration. In this article, they carefully reviewed the progress of multi-way calibration in chromatography, both from the experimental and theoretical perspectives. The organization of data prior to chemometric modelling and the discussion of AFOMs are the highlights of this review. In addition, the structure of this article is clear, and the references are abundant.
Action
We appreciate the reviewer comment.
Comment
However, some revisions as mentioned below should be made before publication to improve its quality.
- The following equations are obviously wrong and must be revised.
(1) In page 10, Eq. 1, the subscript ikj should be ijk.
Action
Fixed
Comment
(2) In page 10, Eq. 2, the equation must be rewritten, because the three-way data array should be the sum of the outer product of each column vector of the matrices A, B and C.
In page Eq. 3, there is the same mistake.
Action
Fixed
The following reference will help you:
Bro R. PARAFAC. Tutorial and applications. Chemom. Intel. Lab. Syst., 1997, 38(2): 149-171.
Comment
(3) In page 11, Eq. 4, the subscript ikj should be ijk. The equation cannot clearly reflect the characteristics of PARAFAC2, so I suggest that the author rewrite the equation according to the original reference.
Action
Our Eq. 4 may be formally different than the original formulation, but it represents the PARAFAC2 model, since bjn(i) is explained to be the jth element of the b profile, which depends on the sample index i. This was explained in more detail in the revised version.
Comment
- In Fig.1 and Fig. 2, we think using double arrows to describe the length of each dimension is not enough, because it is not conducive to the development of algorithms based on graphical representations. It is recommended to use a single arrow to determine the direction.
Action
Fixed
The following reference will help you:
Wu, H.L.; Wang, T.; Yu, R.Q., Recent advances in chemical multi-way calibration with second-order or higher-order advantages: Multilinear models, algorithms, related issues and applications. TrAC Trends Anal. Chem. 2020, 130, 115954.
Comment
- In page 3, lines 110-128, these two paragraphs are not clearly related to "LC multi-way data generation", so they should be placed elsewhere in this article or modified.
Action
We have briefly commented on fluorescence data matrices because they are the best examples to introduce the notions of trilinearity and trilinearity loss. We have reduced the paragraphs in question, but kept them in the place they were, with a comment on the fact that they are being used as examples of data types.
Comment
- For the convenience of readers, the definition of the different “non-multilinear types” should be briefly introduced.
Action
Two comments were added on this issue: one where non-trilinear data are mentioned the first time, and a second one for non-quadrilinear data.
Comment
- The hyperlinks of many references are incorrect, such as “Error! Reference source not found” in line 293. The authors need to check them carefully.
Action
Fixed
Comment
- In page 20, ref. 78, “0.0-536 ng mL-1” should be revised as “0.1-536 ng mL-1”.
Action
Fixed
Reviewer 2 Report
Please see attached document.

Author Response
Comment
- General: The work of Chiappini et al. is a review on the advances and applications in the multi-way calibration field and after minor revisions as they are detailed below, I recommend publication.
- Minor concerns:
- Line 293: :Error! Reference source not found..” Please correct appropriately.
Action
Fixed
Comment
- In Figure 1, please place the labels of y-axis to red horizontally and not as they are.
Action
Fixed
Comment
- In page 8, the scheme is cut. Please correct.
Action
Fixed
Comment
- On page 9, again the labels of axis should be corrected.
Action
Fixed
Comment
- Line 335, same mistake with the reference. Please correct throughout the manuscript.
Action
Fixed